behaviour, ecology

wildlife damage, beehives, *Ursus arctos*, human–wildlife coexistence, spatial risk modelling, multi-scale analysis

**Author for correspondence:**
Carlos Bautista
e-mail: carlos@iop.krakow.pl

# Spatial ecology of conflicts: unravelling patterns of wildlife damage at multiple scales

Carlos Bautista[1], Eloy Revilla[2], Teresa Berezowska-Cnota[1], Néstor Fernández[3,4], Javier Naves[2] and Nuria Selva[1]

[1]Institute of Nature Conservation of the Polish Academy of Sciences (IOP PAN), Adama Mickiewicza 33, 31-120 Kraków, Poland
[2]Estación Biológica de Doñana CSIC (EBD-CSIC), Americo Vespucio 26, 41092 Sevilla, Spain
[3]German Centre for Integrative Biodiversity Research (iDiv), Halle-Jena-Leipzig, Puschstraße 4, 04103 Leipzig, Germany
[4]Institute of Biology, Martin Luther University Halle-Wittenberg, Am Kirchtor 1, 06108 Halle (Saale), Germany

CB, 0000-0003-3979-797X; ER, 0000-0001-5534-5581; TB-C, 0000-0002-2566-6672;
NF, 0000-0002-9645-8571; JN, 0000-0003-3773-0288; NS, 0000-0003-3389-201X

Human encroachment into natural habitats is typically followed by conflicts derived from wildlife damage to agriculture and livestock. Spatial risk modelling is a useful tool to gain the understanding of wildlife damage and mitigate conflicts. Although resource selection is a hierarchical process operating at multiple scales, risk models usually fail to address more than one scale, which can result in the misidentification of the underlying processes. Here, we addressed the multi-scale nature of wildlife damage occurrence by considering ecological and management correlates interacting from household to landscape scales. We studied brown bear (*Ursus arctos*) damage to apiaries in the North-eastern Carpathians as our model system. Using generalized additive models, we found that brown bear tendency to avoid humans and the habitat preferences of bears and beekeepers determine the risk of bear damage at multiple scales. Damage risk at fine scales increased when the broad landscape context also favoured damage. Furthermore, integrated-scale risk maps resulted in more accurate predictions than single-scale models. Our results suggest that principles of resource selection by animals can be used to understand the occurrence of damage and help mitigate conflicts in a proactive and preventive manner.

## 1. Introduction

Conflicts arising from wildlife damage to livestock and agriculture are one of the most urgent and complex challenges for conservationists today [1,2]. To mitigate these conflicts, public administrations often compensate farmers for the economic losses caused by wildlife damage [3] and eventually encourage measures to prevent this damage [4]. However, compensation and prevention programmes systematically neglect landscape heterogeneity in damage risk [5].

An approach to effectively reduce the impact of wildlife damage is to model risk across space [5]. From an ecological perspective, the risk of damage can be described as the probability of the selection of anthropogenic food resources by wild animals. Resource selection is a scale-dependent process, i.e. inference at a broad scale may not adequately explain resource use at a finer scale [6]. For example, the distance to forests may be a strong predictor of livestock predation at intermediate scales but weak at finer ones [7]. In addition, multi-scale resource selection studies show that broader-scale features can constrain selection at finer scales, i.e. fine-scale foraging decisions depend on the spatial heterogeneity of resources at broader scales [7]. That implies the need to integrate inferences across scales to understand the landscape characteristics that determine the probability of resource selection by animals.

Integrating the output of scale-dependent resource selection functions provides the relative probability of selection at a lower scale (e.g. selection of a farm) conditional

## damage risk modelling

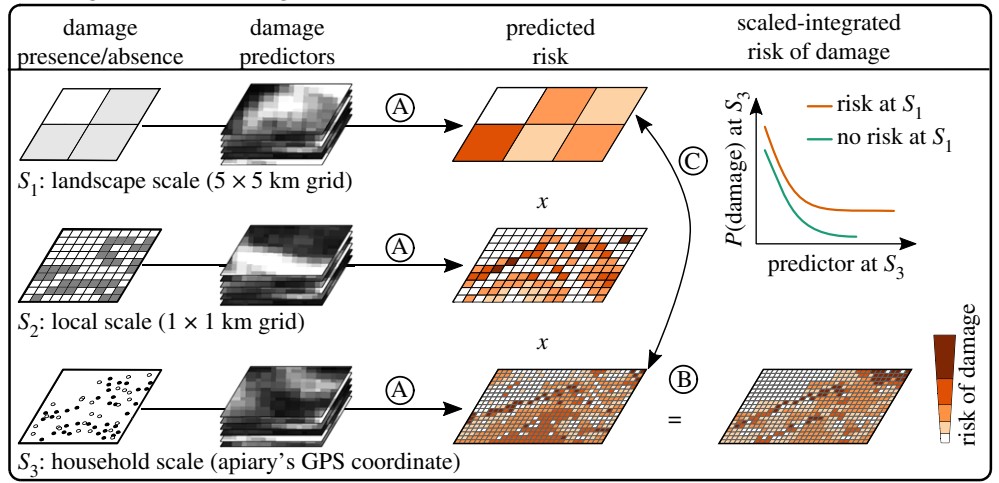

**Figure 1.** Conceptual diagram showing a multi-scale approach to model the risk of wildlife damage. The risk of damage is modelled at multiple scales independently based on *a priori* specified scale-dependent predictions that test one or more general hypotheses. At each scale, the risk of damage can be extrapolated to a larger spatial extension to inform about potential conflict zones in the case of dispersing individual and/or future population increases (A). The resulting predicted probabilities of damage are multiplied at the smallest scale to produce a scale-integrated risk map (B). Finally, it is assessed if the damage risk at fine scale depends on whether the context at larger scales favours damage or not (C). (Online version in colour.)

upon the relative probability of selection at a higher scale (e.g. selection of home range). This is useful for conservation and management because it allows the probability of selection to be predicted and mapped with higher accuracy than single-scale models [8]. In the case of conflict mitigation, public administrations and farmers try to prevent damage at different levels, ranging from the national administrative level to the household [5]. Providing scale-integrated risk maps has potential to understand the ecological processes underlying damage occurrence and providing an effective tool for conflict mitigation.

In this study, we assessed the scale-dependent probability of brown bear (*Ursus arctos*) damage to apiaries in the Polish Carpathian Mountains (figures 1 and 2). The brown bear is the most abundant terrestrial large carnivore in Europe [9]. Its distribution range has been increasing in the last decades in Europe [9] and is expected to continue growing in the near future [10]. Brown bear predation on domestic beehives is widespread, and in some countries (e.g. Poland), it is nearly the only type of human property that bears damage [11].

In the Carpathian population, bears mainly select forest-dominated areas with a low density of roads and human settlements [12,13]. The species sometimes roam in the surroundings of agricultural fields, where they may find natural food resources, such as berries and herbaceous vegetation [14], as well anthropogenic resources like beehives [11]. We hypothesized that bear damage to beehives would mostly occur in areas of high bear habitat suitability with low human influence [13] but with a high availability and accessibility of apiaries.

To evaluate this hypothesis, we modelled the risk of bear damage to beehives at three scales encompassing (i) the scale of a bear home range (hereafter landscape scale); (ii) the habitat selection of bears within their home ranges and the distribution of apiaries at the local scale (hereafter local scale) and (iii) the microhabitat preferences of bears and the preferences of beekeepers in locating their apiaries (hereafter household scale). We fitted one risk model at each scale and integrated the results into a multi-scale risk map. We ran an additional model at the household scale to evaluate to what extent the use of preventive measures decreases the risk of damage. We finally assessed whether the risk of damage follows a spatially

hierarchical structure, in which the broader landscape context can shape bear damage response to household conditions.

## 2. Methods

### (a) Study area

This study covers the Carpathian Mountain range in the Podkarpackie Province, Poland (figure 2). This area is characterized by gentle slopes and low to medium altitude mountains ranging from 199 to 1199 m.a.s.l. The land is mainly covered by forest (62%) and agriculture (32%). Human density averages 44 inhabitants $km^{-2}$, while the average density of roads is 3.2 $km\,km^{-2}$. Honey production is an important economic activity in the area, mostly carried out in domestic exploitations. Many apiaries are unprotected against bear damage. Others are close to buildings or fenced with mesh fence, and only a few of them are well protected with electric fencing (see electronic supplementary material, figure SA1). The average number of beehives per apiary is 17.8 (s.d. = 18.21), ranging from just one to over a hundred (see electronic supplementary material, figure SA1).

### (b) Bear damage data

We compiled data on bear damage to apiaries from official records collected through the damage compensation programme in the Podkarpackie Province by the Regional Directorate for Environmental Protection in Rzeszów. The compensation scheme has been in place since 1999 and includes damage inspection and verification by trained personnel. After a preliminary exploration of the data, we decided to use only data from 2010 to avoid potential omissions and biases associated with limited knowledge by farmers of the compensation scheme at the initial period of programme implementation. Finally, we filtered out records with imprecise or missing location of the attacked beehives.

We obtained data from 406 bear damage events to apiaries from 2010 to 2017. All these records contained geographical information in the form of geographical coordinates (68% records) or the name of the nearest village—the latter were mapped to the village (figure 2).

Damage to apiaries were transferred to 5 × 5 and 1 × 1 km grids for the landscape and the local-scale analyses. At the landscape scale, we used the same 5 × 5 km grid as Fernández *et al.*

Figure 2. Location of the study area showing the apiaries damaged by the brown bear (*Ursus arctos*) in the Northern Carpathian Mountains (SE Poland, Podkarpackie Province) in the period of 2010–2017. (Online version in colour.)

[12], who provided modelled probabilities of bear occurrence based on habitat characteristics in the Northern Carpathians, including our study area. From that grid, we selected 338 cells covering the Carpathian Mountain range within the Podkarpackie Province. At the local scale, we used a grid of 8450 1 km cells nested in the 5 × 5 km grid. Finally, at the household scale, we used the GPS locations of apiaries sampled during 99 days of fieldwork specifically conducted for this study between August 2014 and June 2015 (see the electronic supplementary material, appendix SA1). In addition to these locations, we used data of damaged apiaries from compensation records for the period of 2014–2017, since during these years the damage inspectors systematically collected GPS locations at damage sites. In total, we gathered information from 293 apiaries, of which 123 were damaged.

## (c) Predictors of damage at different scales

We analysed the occurrence of bear damage to apiaries based on scale-specific predictions within the bear range in the Podkarpackie Province (see electronic supplementary material, table SB1). Specifically, we ran one spatial correlation model per each scale, plus an extra model at the household scale to assess the effect of preventive measures on damage risk, i.e. four models: landscape model, local model, household model and preventive model. To delimit the bear range, we selected the 5 × 5 km cells with bear presence based on [12] and on the location of damage events occurring in 2010–2017. We also added cells that had over 40% of forest cover and were adjacent to at least three cells with bear presence to include places where bears could potentially occur but be undetected. This selection resulted in 159 (out of 338) and 3355 (out of 8450) cells of the 5 × 5 and 1 × 1 km grids, respectively. All the apiaries used for the household and the preventive models were located within this selected area. To assess the probability of damage occurrence at each scale, we

classified all cells and apiaries with binary values, with 0 and 1 for undamaged and damaged cells/apiaries, respectively.

At the landscape scale, we expected the probabilities of bear and apiary presence to be inversely correlated, i.e. bears occurring in forested areas with relatively little human influence and apiaries in more altered landscapes dominated by agriculture (see electronic supplementary material, table SB1). For each 5 × 5 km cell, we extracted the probability of bear presence from [12]. We, then, calculated the probability of apiary presence by modelling the location of apiaries recorded during our fieldwork as a response to different environmental and socioeconomic predictors (see electronic supplementary material, appendix SA). Finally, we calculated the damage probability as a function of bear presence probability and apiary presence probability.

At the local scale, we calculated in each 1 × 1 km cell 12 predictors expected to influence the risk of bear damage to beehives (see electronic supplementary material, table SB1). Specifically, we predicted that damage occurrence is directly related to the densities of humans, settlements and roads, to the proportion of agricultural cover and to the length of forest edges, all of which are higher at low altitudes and gentle slopes (see electronic supplementary material, table SB1). We also expected that the above predicted relationships would have nonlinear effects on damage occurrence. For example, we expected damage risk to have a positive relationship with human population density until a certain threshold in which high human densities would deter bears and shift the relationship into negative.

Finally, at the household scale, we predicted that the apiaries that are more exposed to bears are more vulnerable to bear damage, i.e. far from buildings and located within areas of high probability of bear presence. Accordingly, for each apiary, we calculated the probability of bear presence, the distance to the nearest forest patch, the distance to the nearest building, the number of buildings in a radius of 200 m around the apiary and the forest cover in the same 200 m radius. At this scale, we also aimed to assess the influence of preventive measures in damage occurrence. For that, we used a subsample of 151 apiaries (32 of them damaged) for which we had information about the type of measures used to protect apiaries against bear damage. We only considered as preventive measures properly installed and working electric fences (see electronic supplementary material, figure SA1). Other types of fencing, such as wooden or simple wires, were classified as no prevention. Since the immediate surroundings of apiaries may also influence the occurrence of damage (e.g. less damage occurring in apiaries far from the forest and surrounded by buildings), we also expected an interaction effect between the presence of electric fences and the above explained predictors.

## (d) Damage risk models

We used generalized additive models (GAMs) to analyse the occurrence of bear damage to apiaries and predict the probability of damage at the three scales (landscape, local and household) and to assess the effect of preventive measures on damage risk at the household scale (preventive model). We fitted all GAMs with a binomial error distribution and a logarithmic link with damage occurrence (1) versus absence (0) as a response to different environmental and socioeconomic predictors (see above). For the landscape and local models, we used data from the period of 2010–2015 to build the models and data from the period of 2016–2017 to evaluate our predictions. For the household model, we used all available data about apiaries located with GPS in the period of 2014–2017. We used a maximum-likelihood method to estimate smoothing parameters. We added a second penalty in the null space for each smooth term in each model to allow the model to reject the least relevant terms for predictions [15]. To avoid collinearity, we excluded the highly correlated variables through a stepwise procedure based on the variance inflation factor [16]. We only included the predictors with a variance

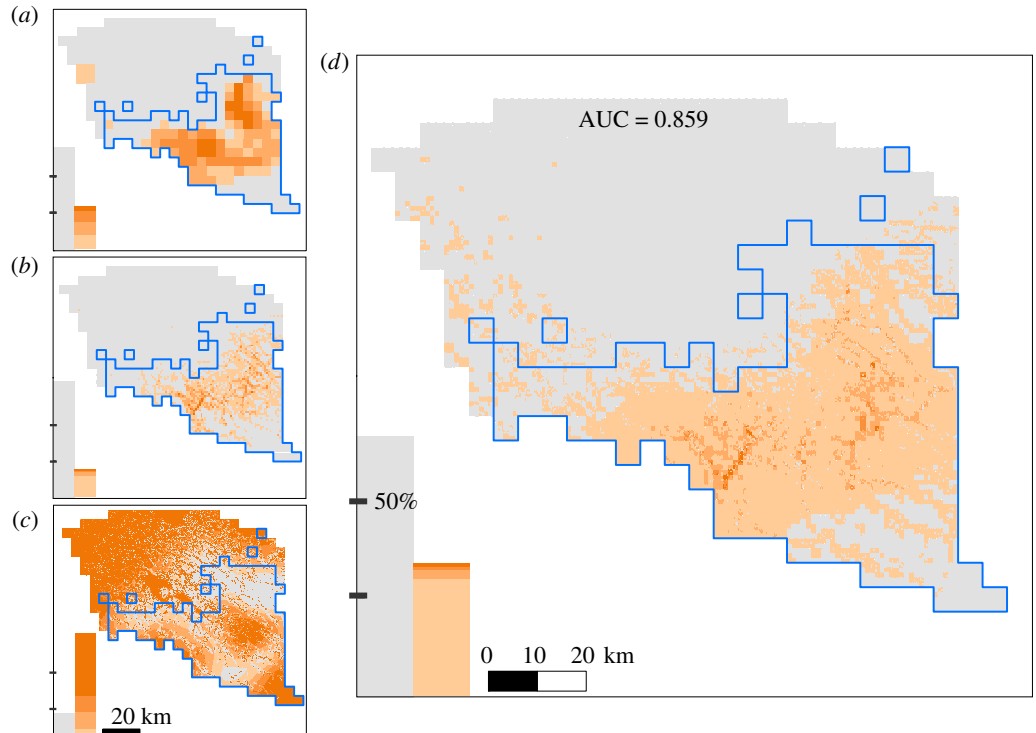

**Figure 3.** Risk maps showing the relative probabilities of brown bear damage to apiaries in the Northern Carpathians (SE Poland) at three scales: 5 × 5 km (*a*), 1 × 1 km (*b*) and 0.25 × 0.25 km (*c*). The relative probability of damage was predicted at each scale based on the coefficients of GAMs run within the bear distribution range (cells delimited by the blue line). That probability was then extrapolated to the potential bear habitat within the Podkarpackie Province to inform about potential conflict zones in the case of future population increases. The relative probabilities of bear damage were multiplied at the smallest scale to produce a scale-integrated risk map (*d*). Predicted risk of damage for all maps was classified using the maximized sum of sensitivity–specificity. The values below the threshold are considered as predicted absence of damage (grey colour). The values above the thresholds were divided into four equal-interval classes of damage risk (the darker the orange colour, the higher the risk). The bar plots at the bottom-left of each panel show the relative frequency of the different risk classes in the map (left bars represent predicted absences and right ones the classes of damage risk). (Online version in colour.)

inflation factor lower than two using a correlation threshold of 0.60. We included in all models an interaction term of the geographical coordinates to account for spatial trends in the data across large geographical distances [17]. For the preventive model, we included the main term 'prevention' (as a categorical linear predictor) plus two smothers for each significant predictor in the household model (one smother for the group 'prevention = yes' and another for 'prevention = no'). This allowed assessing the compounding effect of the presence of preventive methods and the immediate landscape characteristics of the apiaries on damage risk. We run spline correlograms on the occurrence of predation events and on the residuals of all models to assess for the remaining spatial autocorrelation. All statistical analyses were performed in R (v. 3.5.1, R Development Core Team 2018) using the packages mgcv to run GAMs [18], ncf to assess the spatial autocorrelation and mgcViz to visualize the results of GAMs [19].

### (e) Model evaluation

We measured the predictive capacity of the models on damage occurrence using the area under the receiver operating characteristics curves (AUC); the overall rate of correct classifications (accuracy) and the proportions of correctly classified presences (sensitivity) and absences (specificity) of damage to apiaries. For each model, we set the optimal threshold for discriminating damage using the maximized sum of sensitivity and specificity in the receiver operating characteristic curve. For the landscape and local models, we carried out an internal evaluation by computing the performance metrics using data from 2010 to 2015 used to fit the model. We also performed an external evaluation considering the ability of the model to predict bear damages to

apiaries using independent data for the period of 2016–2017. For the household and preventive models, we only performed the internal evaluation because we used all the observations for which we had data on the described predictors to fit them.

### (f) Scale-integrated risk mapping

We extrapolated the risk of bear damage across the Carpathian mountain range within the Podkarpackie Province (figures 1, 2 and 3). We performed this extrapolation beyond the bear distribution area to inform about potential conflict zones in the case of dispersing individual bears and/or future population increases [13]. Specifically, we predicted the risk of damage at each scale based on the coefficients of its corresponding risk model. To extrapolate the risk of damage at the household scale, we divided each of the 8450 cells of 1 km side into 16 cells of 0.25 km side (i.e. 135 200 cells of 0.25 × 0.25 km) and calculated the predictors used in the household model at the centroid of each 0.25 km-side cell.

We integrated the predicted risk of damage across scales at the 0.25 × 0.25 km resolution. To that end, we characterized each 0.25 km-side cell with the probability of damage estimated at each of the three study scales, i.e. three values of damage probability for each cell. We then scaled the predicted probabilities in each cell between 0 and 1 based on the following formula:

$$P(\text{damage}) = \frac{(P(x) - P_{\min})}{(P_{\max} - P_{\min})}.$$

We scaled the probabilities between zero and one to give equal weight to the predicted risk at every scale. Finally, we calculated the scaled-integrated probability of damage to apiaries at each cell by multiplying the damage probabilities at the

**Table 1.** Results from GAMs analysing the occurrence of brown bear damage to apiaries in the Northern Carpathians (SE Poland) at three scales: landscape (5 × 5 km), local (1 × 1 km) and household (apiary's GPS coordinates). The estimated degrees of freedom (*Edf*) for each smooth term are provided. Generally, the higher the Edf the more nonlinear the smoothing spline with Edf = 1 indicating a linear function. However, since we added a second penalty in the null space for each smooth term, Edfs ≤ 1 are not necessarily linear and an Edf near zero indicates that the effect of that smooth term is removed from the model. The smoother effect of the interaction of the geographical coordinates is provided in electronic supplementary material, appendix SB.

| spline fits | Edf | smooth effects |
|---|---|---|
| *landscape model (N = 157, adjusted R$^2$ = 0.224, deviance explained = 21.3%)* | | |
| s(probability of bear presence) | 1.60* | |
| s(probability of apiary presence) | 0.91*** | |
| s(X-coordinate, Y-coordinate) | 5.75** | electronic supplementary material, figure SB2 |
| *local model (N = 3925, adjusted R$^2$ = 0.040, deviance explained = 12.5%)* | | |
| s(slope) | ∼0 | no effect |
| s(agricultural cover) | 0.90 | |
| s(density of major roads) | 1.23** | |
| s(density of minor roads) | ∼0 | no effect |
| s(density of very small roads) | ∼0 | no effect |
| s(forest edge) | 3.36*** | |
| s(X-coordinate, Y-coordinate) | 12.8*** | electronic supplementary material, figure SB3 |
| *household model (N = 293, adjusted R$^2$ = 0.379, deviance explained = 36.4%)* | | |
| s(probability of bear presence) | ∼0 | no effect |
| s(distance to nearest building) | ∼0 | no effect |
| s(distance to nearest forest patch) | 1.28* | |
| s(number of buildings in a 200 m radius) | 1.80*** | |
| s(forest cover in a 200 m radius) | 1.05^ | |
| s(X-coordinate, Y-coordinate) | 16.97*** | electronic supplementary material, figure SB4 |

s = spline; approximate significance of smooth terms based on *p*-values: 0, ***0.001, **0.01, *0.05, ^0.1.
∼0 = values less than 0.1.

landscape, local and household scales following Decesare *et al.* [20] as follows:

$$\text{Scale} - \text{integrated probability of damage} = P(S_1) \times P(S_2) \times P(S_3),$$

where $P(S_1)$, $P(S_2)$ and $P(S_3)$ are the relative probabilities of damage for a given 0.25 km-side cell at the landscape, local and household scales, respectively.

## (g) Assessing whether the landscape context shapes bear damage response to household conditions

We assessed if the damage risk at a fine scale depends on whether the context at larger scales favours damage or not. For that, we first selected the 0.25 km-side cells encompassing damaged and undamaged apiaries (i.e. 272 cells). Then, we characterized the selected cells according to whether they were located within an area predicted as risky or safe in the risk maps at the landscape and local scales (figure 3). As a result, we had four subsets of the 0.25 × 0.25 km grid, comprising apiaries located in (i) risky landscape conditions, (ii) safe landscape conditions, (iii) risky local conditions and (iv) safe local conditions. We used GAMs to predict the probability of damage for each subset of data. We included as predictors the variables previously identified as significant in the household model.

# 3. Results

## (a) Correlates of brown bear damage risk

The results from the landscape model showed that the probability of damage occurrence steadily increased with the probability of apiary presence and, to a lesser extent, with high probabilities of bear presence (table 1 and electronic supplementary material, figure SB2). At the local scale, the damage probability increased with the length of forest edge and with low densities of major roads. It also increased with low values of agricultural cover (table 1 and electronic supplementary material, figure SB3). At the household scale, we found that the risk of damage decreased with an increasing density of buildings in a 200-m radius around the apiaries and increased in apiaries located near forest patches and surrounded by forests (table 1 and electronic supplementary material, figure SB4). Overall, the occurrence of damage had a negative relationship with the distance to the nearest forest patch (see electronic supplementary material, figure SB4). Results from the preventive model showed that apiaries with preventive measures were those with higher risk of being attacked (table 1 and electronic supplementary material, figure SB5). Also, an increasing density of buildings in a 200 m radius decreased the probability of damage in apiaries with no

**Table 2.** Results from a GAM analysing the compounding effect of preventive measures and the surroundings of the apiaries on the occurrence of brown bear damage to apiaries in the Northern Carpathians (SE Poland). The estimated degrees of freedom (*Edf*) for each smooth term are provided. The smoother effect of the interaction of the geographical coordinates is provided in the electronic supplementary material, appendix SB.

| spline fits | Edf | smooth effects |
|---|---|---|
| *prevention model (N = 151, adjusted R² = 0.449, deviance explained = 49.1%)* | | |
| prevention (yes)[a] | 1.70 (± 0.73)* | — |
| s(distance to nearest forest patch): prevention = no | ∼0 | no effect |
| s(distance to nearest forest patch): prevention = yes | ∼0 | no effect |
| s(number of buildings in a 200 m radius): prevention = no | 0.92** | |
| s(number of buildings in a 200 m radius): prevention = yes | ∼0 | no effect |
| s(forest cover in a 200 m radius): prevention = no | ∼0 | no effect |
| s(forest cover in a 200 m radius): prevention = yes | ∼0 | no effect |
| s(X-coordinate, Y-coordinate) | 14.13** | electronic supplementary material, figure SB5 |

[a]Linear fit for which is reported the estimate ± standard error instead of the Edf.

s = spline; approximate significance of smooth terms based on *p*-values: 0, ***0.001, **0.01, *0.05, ^0.1.

∼0 = values less than 0.1.

preventive measures (table 2 and electronic supplementary material, figure SB5).

Risk models showed medium to high predictive accuracy according to the internal evaluation: AUC = 0.79–0.95. The predictive accuracy was lower for the external evaluation: AUC = 0.68–0.63 (see electronic supplementary material, table SB2).

## (b) Scale-integrated risk map

The scale-integrated risk map predicted that 66% of the bear range in the Podkarpackie Province is at some level of risk of bear damage to apiaries, of which 1% is considered to be at high or very high risk (figure 3 and electronic supplementary material, table SB2). The spatial location of high-risk zones within the bear range was consistent across scales. Considering the potential bear habitat within the Podkarpackie Province, 32.7% of the area was at moderate risk, and 0.3% at high risk. The scaled-integrated risk map had a high classification accuracy (AUC = 0.856, figure 3).

## (c) Landscape context can shape bear damage response to household conditions

The predicted risk of damage at the household scale depended on the risk predicted at larger scales. In other words, broad landscape characteristics determined to what extent the immediate surroundings of the apiary influence its vulnerability. Specifically, an apiary surrounded by several buildings and more than 80 m away from the forest edge is up to three times more likely to be damaged by a bear when it is inside (versus outside) a landscape that favours damage (figure 4). To a lesser extent, the probability of damage at the household scale also increased when the environmental characteristics at the local scale favour damage (figure 4).

# 4. Discussion

## (a) Patterns and correlates of damage risk

Our results illustrate that the spatial patterns of bear damage to apiaries are a complex ecological issue modulated by multiple environmental factors and their interactions across several scales. We found that high risk of damage is associated with areas of interface between agricultural landscapes that are suitable for beekeeping (landscape scale) and forest patches that facilitate the movement of bears within their home range [21] (local and household scales). In addition, we found that a high building density in the immediate surroundings of an apiary (household scale) was related to low risk of damage. The overall interpretation of these results confirms, as we hypothesized, that the habitat preferences of bears (to find resources) and beekeepers (to install apiaries) together with the bear's natural tendency to avoid humans determine the risk of bear damage to apiaries at multiple scales.

Our results showed that local-scale patches with a high density of forest edges, roads and agricultural land are susceptible to bear damage. Similar patterns have been observed for other wildlife in different landscapes. For instance, the risk of livestock predation by leopards (*Panthera pardus*) in Bhutan or of crop predation by Asian elephants (*Elephas maximus*) in India also increased in agricultural fields and near roads, respectively [22,23]. Overall, this pattern shows that the risk of damage at medium scales depends directly on the availability and accessibility of farms and crops, both of which are higher in the surroundings of rural human settlements. The conversion of natural ecosystems to agricultural land has steadily increased over the 20th century and is projected to keep increasing globally [24]. As wilderness becomes converted into agricultural land, conflicts arising from damage are also expected to increase [25,26]. Conversely, and in spite of the general trend of agriculture expansion, many regions (e.g. Europe) have experienced a conversion from agriculture into forest habitat mainly as a result of socio-political dynamics like the rural exodus [27]. Land abandonment in rural areas can facilitate conflicts. For example, the decrease and ageing of local population in central Japan has increased the incidence of leaving unattended fruit trees and unharvested crops, which attract Asiatic black bears (*Ursus thibetanus*) and Japanese macaques (*Macaca fuscata*) to villages [28].

Yet, the accessibility of farms and agricultural land (and, thus, the risk of damage) may well be compromised by the landscape characteristics in their most immediate

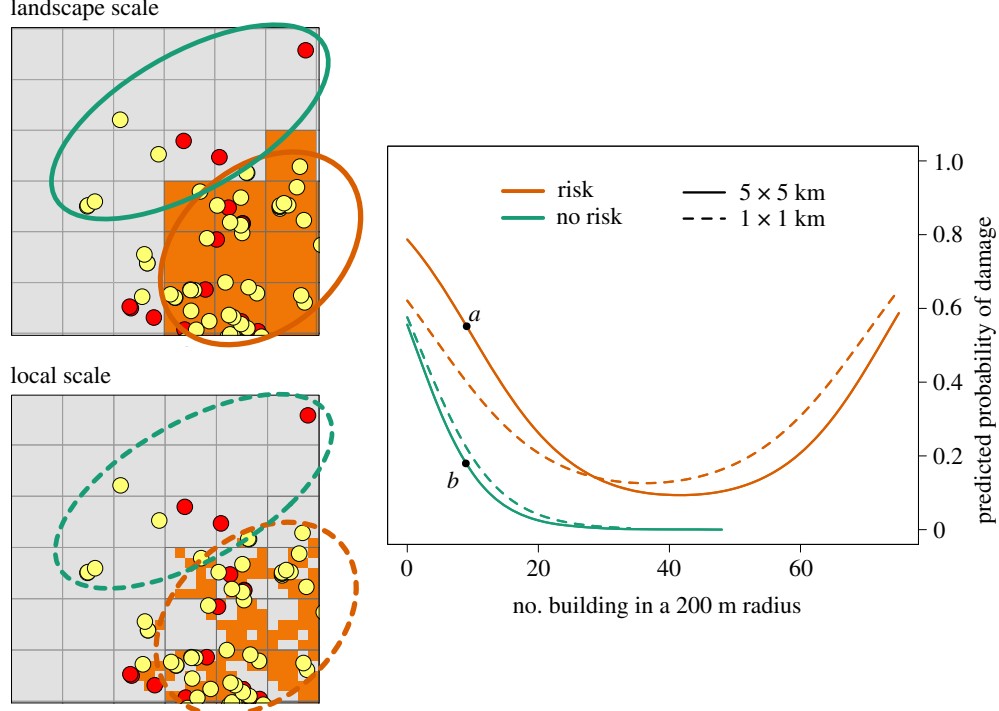

**Figure 4.** Predicted probability of brown bear damage to apiaries as a response to the number of buildings in a 200 m radius around the apiary in the Northern Carpathians (SE Poland). Responses are conditioned to whether the landscape characteristics at large scales favour damage or not. Orange and green lines show the probability of damage in apiaries located in landscapes that favoured (orange cells) and did not favour damage (grey cells), respectively. Solid lines indicate landscape classification (favouring damage or not) at the landscape scale (5 × 5 km), and dashed lines at the local scale (1 × 1 km). Red and yellow dots represent damaged and undamaged apiaries, respectively. The damage probabilities were predicted with average values of the distance to the nearest forest patch, forest cover around the apiary (in a 200 m radius), longitude and latitude. An apiary located in a landscape that favours damage (a) can be up to three times more at risk of being damaged than an apiary located in a safe landscape (b). (Online version in colour.)

surroundings. Results from our household model showed that the risk of damage was at its minimum when apiaries were surrounded by several buildings and located far enough from forest patches (table 1 and figure 4; electronic supplementary material, figure SB4). This evidence that the fear associated with human presence influences the decisions of bears to feed on available energy-rich resources, such as honey and larvae from beehives. Similar patterns have been observed in the use of human-derived foods by other conflict-prone species, such as tigers (*Panthera tigris*) and African elephants (*Loxodonta africana*) [22,29]. This trade-off between energy demands and fear (i.e. using accessible and abundant food resource versus avoiding dangerous situations) has been suggested to shape the spatial ecology and decision-making of wild animals [30] and seems to also shape animal behaviours driving the occurrence of damage. That could explain why the apiaries without electric fences but surrounded by buildings (something relatively common in the study area) tend to have lower damage probability than apiaries with electric fence but installed inside or very close to forest patches and with no buildings around (table 1; electronic supplementary material, figures SB6 and SB7). Although electric fences can be very effective in preventing damage [31], their effectiveness is significantly reduced when they are poorly maintained and they are not reinforced with additional preventive measures [32,33], which is frequent in our study area (see electronic supplementary material, figure SA1). Furthermore, animals with high cognitive abilities, like the brown bear, are known to damage the same farms repeatedly across years and to be able to learn how to skip preventive measures at particular farms [33]. This suggests

that, in the absence of effective prevention, anthropized areas (e.g. urban settlements) can act as a protective shield for farms against wildlife damage.

Although our risk models at the landscape and local scales were accurate in extrapolating the risk of damage to the potential bear habitat within the Podkarpackie Province (AUC ≥ 0.9; electronic supplementary material, figure SB9), they were limited in predicting the presence and absence of damage for the two subsequent years (AUC between 0.65 and 0.68; electronic supplementary material, table SB2). That limitation is likely connected to spatio-temporal variation in missing covariates [34] that can also influence the movement and behaviour of bears (e.g. the availability of food resources; [35]). Indeed, we found that the geographical coordinates used to account for spatial structure in damage patterns in our risk models were significant at every scale (table 1), which can be an indication of missing relevant, spatially structured covariates. For example, the presence of supplementary food provided in natural habitats for wildlife is known to alter the movement behaviour of many animals (including bears in temperate forest ecosystems [36]) and can sometimes increase, instead of decrease, the occurrence of damage [37]. That may be the case when feeding sites, which attract wildlife, and beehives are located close to each other. Other factors that can influence the spatio-temporal patterns of damage occurrence are related to the dispersal movements of juveniles [21] or female bears with cubs seeking human infrastructure to prevent infanticide [38]. Including data on species demography and individual movements can help gain a better understanding on the processes shaping the occurrence of damage, as well as achieving more accurate predictions.

## (b) Integrating damage response to habitat characteristics across scales

By combining the results of risk models across multiple scales, we have demonstrated that the broader landscape context can shape animal responses to the immediate environmental characteristics of a farm. For example, the probability of damage to an apiary greatly increased (up to three times; 0.6 versus 0.2) when it was located in cells predicted at risk at the landscape scale (figure 4). This supports previous findings showing that resource selection at fine scales can be constrained by habitat selected at coarser scales [8]. For example, Lipsey *et al.* [8] demonstrated that the fine-scale probability to select grasslands by Sprague's Pipit (*Anthus spragueii*) increased with the proportion of grass at broader landscapes. These conditional relationships in resource selection among scales are rarely tested in risk mapping, and yet, help to gain a more integrative understanding of how animals select different types of resources and are prone to conflict.

To the best of our knowledge, this is the first study integrating scale-dependent responses of animals in the use of farm products. The majority of damage risk assessments to date are based on scale-specific models [5]. Just a few studies have assessed the probability of damage at multiple scales by identifying scale-dependent patterns of livestock predation [22] or the best grain to improve damage predictions [39]. Here, we showed that combining the extrapolations from single-scale risk models into an integrated-scale risk map greatly improved the spatial prediction accuracy (figure 3; electronic supplementary material, figure SB9 and table SB7) and overcame the limitations of single-scale risk mapping on predicting conflicts from other time lapses (see electronic supplementary material, figure SB10). Previous studies integrating the scale-dependent response of animals to the availability of natural resources also resulted in more accurate predictions than traditional, scale-specific models [8,20]. Our study adds evidence that scale integration can be applied to the particular case of wildlife damage to farm products to predict more accurately where conflicts are more likely to occur.

The recommendations derived from scale-integrated risk maps can avoid wasting resources in management actions based on inaccurate recommendations from scale-specific risk maps. For example, the map based on the landscape model wrongly identified a small region in the northwest part of our study area as a priority for conflict mitigation (figure 3). The northwest has, in fact, an optimal habitat to install apiaries (see electronic supplementary material, figure SA4); however, it is relatively far from the current bear distribution and its local context does not favour damage (see local-scale risk map in figure 3). Accordingly, the joint probability of damage rescaled to the landscape scale (see electronic supplementary material, figures SB9 and SB10) reduced by 75% the area identified as at risk in the northwest. Furthermore, rescaling the risk of damage from the integrated risk map at the finest scale to the broadest landscape scale increased the prediction sensitivity in comparison with the predictions derived from the single-scale map (i.e. 90%—versus 82%—of damage locations were identified correctly, see electronic supplementary material, figure SB10). Although the management and decisions on conflict mitigation strategies are taken on broad scales, these scales do not accurately reflect the spatial heterogeneity of damage occurrence [5,40]. Summarizing the results from scale-integrated risk maps from fine to large scales can help to avoid mismatches between the scales of inference and management action and thus, provide better information to managers and policymakers for damage prevention [41].

## (c) Implications for conservation

Proactive and preventive approaches to mitigate conflicts arising from wildlife damage are proved to be more successful over time than reactive approaches [31]. Yet, most efforts invested in conflict mitigation around the world are allocated to reactive approaches (e.g. compensation programmes), thus, compromising the real success of conflict mitigation actions [3,4]. Given that resources for conflict mitigation are usually limited, prioritizing the areas in the landscape and the particular farms that should be protected first would be highly beneficial for damage prevention. Our multi-scale approach allows identifying risk areas on the broad landscape context and, in those areas, selecting the most vulnerable households in which to subsidize preventive measures. Following our case study, beekeepers working in landscapes that favour damage could reduce the probability of experiencing bear damage by more than threefold if they would locate their beehives at least 300 m away from the forest patches and in the vicinity of several buildings (figure 4 and electronic supplementary material, figure SB4). We believe that our approach may be used as a guideline for future damage risk assessments of other wildlife species and in other parts of the world and, thus, help to effectively reduce damage occurrence and enhance human–wildlife coexistence.

Conflicts arising from wildlife damage are predicted to grow due to the recovery and expansion of some wild animal populations into human-dominated landscapes [9] and due to the increasing transformation of natural areas into agriculture fields [24–26]. The use of agriculture lands and suburban areas by wild animals can become an ecological trap for them, impacting species demography and even leading to local extinctions [42]. Indeed, conflict with humans is one of the main threats for the survival of many species of large carnivores and herbivores [25,26]. That is worrying because they play an essential role in ecosystem functioning worldwide [43]. Unfortunately, given the current human population growth, stopping agriculture expansion into natural areas and the conflicts arising from it may be an unrealistic short-term goal [44]. However, using risk models to predict where damage is more likely to occur and have a proactive and preventive attitude towards conflicts is something that farmers, conservation practitioners and policymakers can start doing today.

Data accessibility. All data and code used for the analyses are available from the Dryad Digital Repository: https://doi.org/10.5061/dryad.rfj6q57bc [45].

The data are provided in electronic supplementary material [46].

Authors' contributions. C.B.: conceptualization, data curation, formal analysis, funding acquisition, investigation, methodology, visualization, writing-original draft, writing-review & editing; E.R.: conceptualization, methodology, supervision, writing-review & editing; T.B.-C.: data curation, writing-review & editing; N.F.: conceptualization, methodology, supervision, writing-review & editing; J.N.: conceptualization, writing-review & editing; N.S.: conceptualization, funding acquisition, methodology, project administration, supervision, writing-original draft, writing-review & editing.

All authors gave final approval for publication and agreed to be held accountable for the work performed therein.

Competing interests. We have no competing interests.

**Funding.** This study was funded by the National Science Centre in Poland under agreement nos. UMO-2013/08/M/NZ9/00469, UMO-2017/25/N/NZ8/02861 and UMO-2020/36/T/NZ8/00571. E.R., N.S. and J.N. were supported by project CGL2017- 83045-R AEI/FEDER EU, by the Agencia Estatal de Investigación from the Ministry of Economy, Industry and Competitiveness, Spain, co-financed with FEDER Fundus.

**Acknowledgements.** We thank the Regional Direction of Environmental Protection in Rzeszow, Fundacja Dziedzictwo Przyrodnicze, Robert Gatzka, Aida Parres and volunteers from the Carpathian Brown Bear Project for helping in data collection. We thank the two anonymous referees for their constructive comments.

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
