## [Peer Review File · Proceedings of the Royal Society B: Biological Sciences]

Review History

RSPB-2020-2704.R0 (Original submission)

Review form: Reviewer 1

Recommendation

Accept with minor revision (please list in comments)

Scientific importance: Is the manuscript an original and important contribution to its field?

Good

General interest: Is the paper of sufficient general interest?

Good

Quality of the paper: Is the overall quality of the paper suitable?

Good

Is the length of the paper justified?

Yes

Should the paper be seen by a specialist statistical reviewer?

No

Do you have any concerns about statistical analyses in this paper? If so, please specify them explicitly in your report.

No

It is a condition of publication that authors make their supporting data, code and materials available - either as supplementary material or hosted in an external repository. Please rate, if applicable, the supporting data on the following criteria.

Is it accessible?

Yes

Is it clear?

Yes

Is it adequate?

Yes

Do you have any ethical concerns with this paper?

No

Comments to the Author

This was an interesting paper to read, and relates to the broader field of human-wildlife conflict. A very nice study, kudos! The multi-scale approach is novel, study limitations are acknowledged, and most importantly, the results while largely not unexpected, allow for managers to provide specific recommendations to avoid/minimise bears raiding apiaries. This is great to see as it will presumably be used to promote improved coexistence between humans and bears (a challenge in many parts of the world where bears are reoccupying their former distributions).

I don't have any substantial recommended changes aside from that in places the writing would benefit from some additional proofreading and editing.

Review form: Reviewer 2

Recommendation

Major revision is needed (please make suggestions in comments)

Scientific importance: Is the manuscript an original and important contribution to its field?

Good

General interest: Is the paper of sufficient general interest?

Acceptable

Quality of the paper: Is the overall quality of the paper suitable?

Good

Is the length of the paper justified?

Yes

Should the paper be seen by a specialist statistical reviewer?

No

Do you have any concerns about statistical analyses in this paper? If so, please specify them explicitly in your report.

Yes

It is a condition of publication that authors make their supporting data, code and materials available - either as supplementary material or hosted in an external repository. Please rate, if applicable, the supporting data on the following criteria.

Is it accessible?

Yes

Is it clear?

Yes

Is it adequate?

Yes

Do you have any ethical concerns with this paper?

No

Comments to the Author

I have read with great interest the manuscript by Bautista et al. titled "The spatial ecology of conflicts: Unraveling patterns of wildlife damage to livestock at multiple nested scales". The Authors address an important aspect of predicting human-wildlife conflict spatially by integrating maps of different scales into all-in-one single layers. They found that risk of damage to apiaries by brown bear in the Northern Carpathians was best predicted by integrating broad, landscape scale bear and apiaries presence probability with local scale landscape composition (length of forest edge, road density and percent cropland), and with fine (household) scale building density and distance to forest. Importantly, multiscale risk maps predicted risk more accurately than single-scale maps.

I find the paper generally well written and clear, but I have, however, identified a few major caveats regarding the spatial projection of risk that should be addressed before the paper could be published. I give some more specific comments below that I hope are constructive and will help the Authors in improving their manuscript.

SPECIFIC COMMENTS to the Authors

L210 in my opinion, you should model risk using apiaries without protection only – then you can actually test if the protected one in risky pixels were efficiently protected (ie. decreased damage). This would also inform where to place protection measures in the most risky pixels, and not "waste" these efforts in non-risky pixels. To do so, I suggest that you remove the 151 prevented apiaries for the 293 model (or at least remove the electrified ones that you classify as prevented) and re-run the model. I don't expect large differences, but this would really be the risk based on bear habitat and apiaries placement (regardless of prevention) and then model the effect of prevention in decreasing risk (or not) in the most risky pixels...

L212-218 The approach to scale integration needs to be clarified. It is not clear how you combined the different normalized layers ("based on map overlaying" L 217-218 is unclear). See eg. deCesare et al. 2012, Pitman et al. 2017, Fattedbert et al. 2018 for recent examples of scale integration in different taxa.

L218-220, Figure 1 and Figure 3: I really wonder what is the benefit of resampling the scale integrated layer at the local and landscape coarser resolution. I would imagine that it is most beneficial for the design of management and conservation intervention to work with a single, scale-integrated map that account for drivers at broad and fine scale. Resampling at coarse resolution defeats the idea of scale integration, in my opinion. I do understand that the output are different from the corresponding single-scale maps, but I would question their benefit for applications.

Looking at figure 3, I wonder why you projected risk beyond bear range: you need to justify this or then limit your spatial protection within the blue polygon (Fig. 3). My intuition is that risk should be trivially zero (at all 3 scales) if no bear are present... Unless you consider the risk map

as a habitat suitability model and wish to map potential risk if bear were present/when they recolonize, which could inform prevention measures ahead of the recolonization by bear. If this the case, this point needs to be clarified.

Also, it is unclear if you have apiaries distribution data beyond the bear range area to be able to project risk over there. You would need apiaries data to be able to inform the grid cells with values of the distance to forest.

L223-224 Unclear

LITERATURE CITED

DeCesare, N. J., Hebblewhite, M., Schmiegelow, F., Hervieux, D., McDermid, G. J., Neufeld, L., ... & Wheatley, M. (2012). Transcending scale dependence in identifying habitat with resource selection functions. *Ecological Applications*, 22(4), 1068-1083.

Fattebert, J., Michel, V., Scherler, P., Naef-Daenzer, B., Milanese, P., & Gruebler, M. U. (2018). Little owls in big landscapes: Informing conservation using multi-level resource selection functions. *Biological Conservation*, 228, 1-9.

Pitman, R. T., Fattebert, J., Williams, S. T., Williams, K. S., Hill, R. A., Hunter, L. T., ... & Balme, G. A. (2017). Cats, connectivity and conservation: incorporating data sets and integrating scales for wildlife management. *Journal of Applied Ecology*, 54(6), 1687-1698.

Decision letter (RSPB-2020-2704.R0)

30-Dec-2020

Dear Mr Bautista:

I am writing to inform you that your manuscript RSPB-2020-2704 entitled "The spatial ecology of conflicts: Unraveling patterns of wildlife damage to livestock at multiple nested scales" has, in its current form, been rejected for publication in *Proceedings B*.

This action has been taken on the advice of referees, who have recommended that substantial revisions are necessary. With this in mind we would be happy to consider a resubmission, provided the comments of the referees are fully addressed. However please note that this is not a provisional acceptance.

Sincerely,
Dr Locke Rowe
mailto: proceedingsb@royalsociety.org

Associate Editor
Board Member: 1

Comments to Author:

Both reviewers were positive about this study of the scale-dependence of bear damage to apiaries as an example of human-wildlife conflict. Reviewer 2 raises important caveats about the risk modelling which will require reanalysis. In terms of the presentation of the article, I strongly encourage the authors to present as clearly as possible the broader importance or biological implications of the research in the context of the scale-dependence of factors influencing human-wildlife conflict: at present the Discussion is very largely focused on the bee/bear case study, and for a general biology journal the Discussion should relate this research more clearly to the general context, for example to the material discussed and cited near the start of the Introduction (e.g. references 4-12).

Reviewer(s)' Comments to Author:

Referee: 1

Comments to the Author(s)

This was an interesting paper to read, and relates to the broader field of human-wildlife conflict. A very nice study, kudos! The multi-scale approach is novel, study limitations are acknowledged, and most importantly, the results while largely not unexpected, allow for managers to provide specific recommendations to avoid/minimise bears raiding apiaries. This is great to see as it will presumably be used to promote improved coexistence between humans and bears (a challenge in many parts of the world where bears are reoccupying their former distributions).

I don't have any substantial recommended changes aside from that in places the writing would benefit from some additional proofreading and editing.

Referee: 2

Comments to the Author(s)

I have read with great interest the manuscript by Bautista et al. titled "The spatial ecology of conflicts: Unraveling patterns of wildlife damage to livestock at multiple nested scales". The Authors address an important aspect of predicting human-wildlife conflict spatially by integrating maps of different scales into all-in-one single layers. They found that risk of damage to apiaries by brown bear in the Northern Carpathians was best predicted by integrating broad, landscape scale bear and apiaries presence probability with local scale landscape composition (length of forest edge, road density and percent cropland), and with fine (household) scale building density and distance to forest. Importantly, multiscale risk maps predicted risk more accurately than single-scale maps.

I find the paper generally well written and clear, but I have, however, identified a few major caveats regarding the spatial projection of risk that should be addressed before the paper could be published. I give some more specific comments below that I hope are constructive and will help the Authors in improving their manuscript.

SPECIFIC COMMENTS to the Authors

L210 in my opinion, you should model risk using apiaries without protection only – then you can actually test if the protected one in risky pixels were efficiently protected (ie. decreased damage). This would also inform where to place protection measures in the most risky pixels, and not

“waste” these efforts in non-risky pixels. To do so, I suggest that you remove the 151 prevented apiaries for the 293 model (or at least remove the electrified ones that you classify as prevented) and re-run the model. I don’t expect large differences, but this would really be the risk based on bear habitat and apiaries placement (regardless of prevention) and then model the effect of prevention in decreasing risk (or not) in the most risky pixels...

L212-218 The approach to scale integration needs to be clarified. It is not clear how you combined the different normalized layers (“based on map overlaying” L 217-218 is unclear). See eg. deCesare et al. 2012, Pitman et al. 2017, Fattebert et al. 2018 for recent examples of scale integration in different taxa.

L218-220, Figure 1 and Figure 3: I really wonder what is the benefit of resampling the scale integrated layer at the local and landscape coarser resolution. I would imagine that it is most beneficial for the design of management and conservation intervention to work with a single, scale-integrated map that account for drivers at broad and fine scale. Resampling at coarse resolution defeats the idea of scale integration, in my opinion. I do understand that the output are different from the corresponding single-scale maps, but I would question their benefit for applications.

Looking at figure 3, I wonder why you projected risk beyond bear range: you need to justify this or then limit your spatial protection within the blue polygon (Fig. 3). My intuition is that risk should be trivially zero (at all 3 scales) if no bear are present... Unless you consider the risk map as a habitat suitability model and wish to map potential risk if bear were present/when they recolonize, which could inform prevention measures ahead of the recolonization by bear. If this the case, this point needs to be clarified.

Also, it is unclear if you have apiaries distribution data beyond the bear range area to be able to project risk over there. You would need apiaries data to be able to inform the grid cells with values of the distance to forest.

L223-224 Unclear

LITERATURE CITED

DeCesare, N. J., Hebblewhite, M., Schmiegelow, F., Hervieux, D., McDermid, G. J., Neufeld, L., ... & Wheatley, M. (2012). Transcending scale dependence in identifying habitat with resource selection functions. *Ecological Applications*, 22(4), 1068-1083.

Fattebert, J., Michel, V., Scherler, P., Naef-Daenzer, B., Milanese, P., & Gruebler, M. U. (2018).

Little owls in big landscapes: Informing conservation using multi-level resource selection functions. *Biological Conservation*, 228, 1-9.

Pitman, R. T., Fattebert, J., Williams, S. T., Williams, K. S., Hill, R. A., Hunter, L. T., ... & Balme, G. A. (2017). Cats, connectivity and conservation: incorporating data sets and integrating scales for wildlife management. *Journal of Applied Ecology*, 54(6), 1687-1698.

Author's Response to Decision Letter for (RSPB-2020-2704.R0)

See Appendix A.

RSPB-2021-1394.R0

Review form: Reviewer 2

Recommendation

Accept with minor revision (please list in comments)

Scientific importance: Is the manuscript an original and important contribution to its field?

Good

General interest: Is the paper of sufficient general interest?

Good

Quality of the paper: Is the overall quality of the paper suitable?

Good

Is the length of the paper justified?

Yes

Should the paper be seen by a specialist statistical reviewer?

No

Do you have any concerns about statistical analyses in this paper? If so, please specify them explicitly in your report.

No

It is a condition of publication that authors make their supporting data, code and materials available - either as supplementary material or hosted in an external repository. Please rate, if applicable, the supporting data on the following criteria.

Is it accessible?

Yes

Is it clear?

Yes

Is it adequate?

Yes

Do you have any ethical concerns with this paper?

No

Comments to the Author

I have read the revised version of the manuscript by Bautista et al. titled "The spatial ecology of conflicts: Unraveling patterns of wildlife damage at multiple scales". I find this revised version of the manuscript much clearer than the previous one. I commend the Authors for their thorough revision. In particular, I appreciate the effort in exploring my suggestion for modelling risk of apiaries at the household level without using the protected ones in the training data and the detailed presentation of the results in their rebuttal.

I have only a few minor comments remaining (line numbers correspond to the unmarked revised version):

L72 you should clearly spell out what risk you are talking about here, before you can use the term 'risk' alone: "To evaluate this hypothesis, we used modelled the risk of beehives to be damaged by bears at three scales encompassing..."

L76 you should mention here the other model at this scale, too (i.e., the preventive model) - you have done more than these 3 models.

L164 I suggest that you move this entire paragraph earlier in the Methods, and before the predictors' section (L116) - you might have to move the rest of the model section (L150), too. This should inform the reader about what is attempted generally, before diving into the details of each scale. Also, one thing that remains unclear, is how the 123 damaged vs the 170 non-damaged occurrences are treated as a response variable in your GAMs - you need to explicitly tell a relative naïve reader how you have coded them, and which are 1's and which are 0's.

L216 you need at least one reference to back up this approach, e.g. DeCesare et al. 2012. I suggest that you rephrase as: "...by multiplying the damage probabilities at the landscape, local and household scales, following DeCesare et al. 2012:"

LITERATURE CITED

DeCesare, N. J., Hebblewhite, M., Schmiegelow, F., Hervieux, D., McDermid, G. J., Neufeld, L., ... & Wheatley, M. (2012). Transcending scale dependence in identifying habitat with resource selection functions. *Ecological Applications*, 22(4), 1068-1083.

Decision letter (RSPB-2021-1394.R0)

26-Jul-2021

Dear Mr Bautista

I am pleased to inform you that your manuscript RSPB-2021-1394 entitled "The spatial ecology of conflicts: Unraveling patterns of wildlife damage at multiple scales" has been accepted for publication in *Proceedings B*.

The referee(s) have recommended publication, but also suggest some minor revisions to your manuscript. Therefore, I invite you to respond to the referee(s)' comments and revise your manuscript. Because the schedule for publication is very tight, it is a condition of publication that you submit the revised version of your manuscript within 7 days. If you do not think you will be able to meet this date please let us know.

Online supplementary material will also carry the title and description provided during submission, so please ensure these are accurate and informative. Note that the Royal Society will

not edit or typeset supplementary material and it will be hosted as provided. Please ensure that the supplementary material includes the paper details (authors, title, journal name, article DOI). Your article DOI will be 10.1098/rspb.[paper ID in form xxxx.xxxx e.g. 10.1098/rspb.2016.0049].

Sincerely,

Dr Locke Rowe

Associate Editor

Board Member

Comments to Author:

The referee commended the authors on the thorough revision to the manuscript, and I also find that the new text and presentation are greatly improved. Please see the remaining comments of the referee, which refer principally to presentation.

Reviewer(s)' Comments to Author:

Referee: 2

Comments to the Author(s).

I have read the revised version of the manuscript by Bautista et al. titled "The spatial ecology of conflicts: Unraveling patterns of wildlife damage at multiple scales". I find this revised version of

the manuscript much clearer than the previous one. I commend the Authors for their thorough revision. In particular, I appreciate the effort in exploring my suggestion for modelling risk of apiaries at the household level without using the protected ones in the training data and the detailed presentation of the results in their rebuttal.

I have only a few minor comments remaining (line numbers correspond to the unmarked revised version):

L72 you should clearly spell out what risk you are talking about here, before you can use the term 'risk' alone: "To evaluate this hypothesis, we used modelled the risk of beehives to be damaged by bears at three scales encompassing..."

L76 you should mention here the other model at this scale, too (i.e., the preventive model) – you have done more than these 3 models.

L164 I suggest that you move this entire paragraph earlier in the Methods, and before the predictors' section (L116) – you might have to move the rest of the model section (L150), too. This should inform the reader about what is attempted generally, before diving into the details of each scale. Also, one thing that remains unclear, is how the 123 damaged vs the 170 non-damaged occurrences are treated as a response variable in your GAMs – you need to explicitly tell a relative naïve reader how you have coded them, and which are 1's and which are 0's.

L216 you need at least one reference to back up this approach, e.g. DeCesare et al. 2012. I suggest that you rephrase as: "...by multiplying the damage probabilities at the landscape, local and household scales, following DeCesare et al. 2012:"

LITERATURE CITED

DeCesare, N. J., Hebblewhite, M., Schmiegelow, F., Hervieux, D., McDermid, G. J., Neufeld, L., ... & Wheatley, M. (2012). Transcending scale dependence in identifying habitat with resource selection functions. *Ecological Applications*, 22(4), 1068-1083.

Author's Response to Decision Letter for (RSPB-2021-1394.R0)

See Appendix B.

Decision letter (RSPB-2021-1394.R1)

04-Aug-2021

Dear Mr Bautista

I am pleased to inform you that your manuscript entitled "The spatial ecology of conflicts: Unraveling patterns of wildlife damage at multiple scales" has been accepted for publication in *Proceedings B*.

Data Accessibility section

Open Access

Paper charges

Sincerely,

Appendix A

Response to Referees on the revised paper

“The spatial ecology of conflicts:

Unraveling patterns of wildlife damage at multiple scales”

(Ref: RSPB-2020-2704)

**Submitted to:
Proceedings of the Royal Society B**

Dear Editor,

Many thanks for your recent letter inviting us to resubmit our paper *The spatial ecology of conflicts: Unraveling patterns of wildlife damage at multiple scales* (RSPB-2020-2704). We really appreciate the opportunity to resubmit the paper, as well as your time and the time of the two Referees in reading the manuscript and providing constructive comments. We believe that they have largely helped us to improve the manuscript.

In summary, following the suggestions by the Editor and Referees, we have (1) removed emphasis from applied conservation issues and our particular case study, and (2) have framed different parts and elements of the manuscript using resource selection theory and elaborating on the ecological implications of our results.

Thank you very much for your time and consideration.

Response to Referees

(PLEASE NOTE THAT LINE NUMBERS REFER TO THE REVISED MANUSCRIPT WITH CHANGES MARKED)

Associate Editor

Both reviewers were positive about this study of the scale-dependence of bear damage to apiaries as an example of human-wildlife conflict. Reviewer 2 raises important caveats about the risk modelling which will require reanalysis. In terms of the presentation of the article, I strongly encourage the authors to present as clearly as possible the broader importance or biological implications of the research in the context of the scale-dependence of factors influencing human-wildlife conflict: at present the Discussion is very largely focused on the bee/bear case study, and for a general biology journal the Discussion should relate this research more clearly to the general context, for example to the material discussed and cited near the start of the Introduction (e.g. references 4-12).

RESPONSE: We appreciate this comment. We have made substantial changes in the Discussion to frame our study in a more general context in terms scale dependency and the determinants of human-wildlife conflicts. For that, we have included new references illustrating the spatial complexity of resource selection (including the use of farm products) by other taxa and in other parts of the globe. We have also modified the Introduction (lines 44-71), some parts of the Results (lines 302-320 and 335-351) and the title for consistency with this broader focus.

Referee 1

This was an interesting paper to read, and relates to the broader field of human-wildlife conflict. A very nice study, kudos! The multi-scale approach is novel, study limitations are acknowledged, and most importantly, the results while largely not unexpected, allow for managers to provide specific recommendations to avoid/minimise bears raiding apiaries. This is great to see as it will presumably be used to promote improved coexistence between humans and bears (a challenge in many parts of the world where bears are reoccupying their former distributions).

I don't have any substantial recommended changes aside from that in places the writing would benefit from some additional proofreading and editing.

RESPONSE: Thanks for the positive comment. We have proofread the manuscript to polish the writing and correct language-related mistakes.

Referee 2

I have read with great interest the manuscript by Bautista et al. titled "The spatial ecology of conflicts: Unraveling patterns of wildlife damage to livestock at multiple nested scales". The Authors address an important aspect of predicting human-wildlife conflict spatially by integrating maps of different scales into all-in-one single layers. They found that risk of damage to apiaries by brown bear in the Northern Carpathians was best predicted by integrating broad, landscape scale bear and apiaries presence probability with local scale landscape composition (length of forest edge, road density and percent cropland), and with fine (household) scale building density and distance to forest. Importantly, multiscale risk maps predicted risk more

accurately than single-scale maps.

I find the paper generally well written and clear, but I have, however, identified a few major caveats regarding the spatial projection of risk that should be addressed before the paper could be published. I give some more specific comments below that I hope are constructive and will help the Authors in improving their manuscript.

in my opinion, you should model risk using apiaries without protection only – then you can actually test if the protected one in risky pixels were efficiently protected (ie. decreased damage). This would also inform where to place protection measures in the most risky pixels, and not “waste” these efforts in non-risky pixels. To do so, I suggest that you remove the 151 prevented apiaries for the 293 model (or at least remove the electrified ones that you classify as prevented) and re-run the model. I don’t expect large differences, but this would really be the risk based on bear habitat and apiaries placement (regardless of prevention) and then model the effect of prevention in decreasing risk (or not) in the most risky pixels...

RESPONSE: Thank you for raising this point. Ideally we would have included the effect of prevention, among other predictors, in one single model using the full dataset to assess the spatial risk of damage at the household scale. However, the structure of the data did not allow for such analyses. Our dataset consisted of 293 apiaries with information about the occurrence of bear damage, 151 of them with information about the use of preventive measures (only 24 protected with electric fences plus 127 unprotected), whereas the remaining 142 had no information about preventive measures. In the previous version of the manuscript we did run two complementary models: a first one using the full dataset and assessing the risk of damage in relation to landscape features in the immediate surroundings of the apiaries (household model) and a second one to evaluate to what extent the use of preventive measures decreases the risk of damage (preventive model). By using all available data we maximized the statistical power of our analysis and allowed our models to describe, in a reliable way, the mechanisms driving damage occurrence in our study area.

The Referee proposed a first model using only not prevented apiaries (i.e. 127 observations) and then to model the effect of prevention only in risky pixels (ca. 60 observation). This analysis implies excluding a relatively large number of observations, and potentially missing some patterns that could be revealed from our data (see below).

We run the analyses proposed by the Referee and compared the results with ours in terms of predictive accuracy and how well they describe the association between preventive measures and the probability of damage in our study area. Specifically, we used generalized additive models to analyze the occurrence of damage in apiaries that lacked preventive measures as response of the distance to the nearest building, and the forest cover and building density in a 200-meters radius around the apiary (hereafter the *Referee’s household model*). Other predictors were excluded to avoid collinearity. We followed the same exact steps for the statistic analysis as we did for the other risk models (see detail explanation in the section 2(d) of the main manuscript –lines 201 to 236-). Based on the coefficients of the model we predicted the probability of damage occurrence on the full sample of apiaries. To categorize the risk of damage we set the optimal threshold for predicted absence versus presence of damage using the maximized sum of sensitivity and specificity in the receiver operating characteristic (ROC) curve. We considered the values below the threshold as very low risk of damage and divided the values above the threshold into two equal-interval classes representing moderate and high risks of damage. We classified the apiaries within each category of risk (very low, moderate and high) based on the occurrence of damage (yes/no) (see Figure 1 below). Finally we modeled the probability of damage in apiaries classified as risky (moderate and high risk of damage) as response of prevention in interaction with the

density of building in a 200-meters radius; i.e., a categorical linear predictor for the term prevention plus one smother for the group ‘prevention=yes’ and another for ‘prevention=no’ (hereafter *Referee’s preventive model*).

Overall the predictive accuracy of the *Referee’s household model* was smaller than that of our original household model; AUC = 0.786 vs 0.882. The *Referee’s household model* classified satisfactorily the occurrence of damage in unprotected apiaries across the three predicted categories of risk; all unprotected apiaries classified as very low risk had not been previously damaged and almost all those classified as in high risk had experienced damage (Figure 1 below). Yet, its classification for the apiaries that had preventive measures and for those for which information about preventive measure was unavailable was worse than that of our original household model (see Figure 1 below).

Figure 1. The number of damaged and undamaged apiaries within the bear range in Northern Carpathians, SE Poland. The apiaries are grouped based on the presence and absence of measures to prevent bear damage and classified in different categories of predicted risk of damage. The categories of predicted risk of damage were calculated based on the coefficients of Generalized Additive Models. The apiaries were classified twice in the same categories of damage risk: first based on the coefficients of the household model explained in the main document and second based on the coefficients resulting from the model proposed by the Referee.

The results from the Referee’s preventive model did not show any effect of prevention in the occurrence of damage (see Table 1 below). This lack of pattern is likely related to the fact that preventive measures in our study area are mostly installed in apiaries located in risky and remote areas (see Figure 2 below). In fact, our results captured that apiaries with preventive measures were those with higher risk of being attacked (lines 315-318). These results are also connected to the fact that preventive measures in our study area are often ineffective in preventing damage as we explain in the Discussion section (lines 404 -417).

Table 1. Results from generalized additive models analyzing the effect of preventive measures on the occurrence of brown bear predation on beehives in apiaries considered to be in risk of predation. The estimated degrees of freedom (*Edf*) for each smooth term are provided. Generally, the higher the *Edf*, is the more non-linear is the smoothing spline with *Edf* =1 indicating a linear function. However, since we added a second penalty in the null space for each smooth term, *Edfs* ≤ 1 are not necessarily linear and an *Edf* near zero indicates that the effect of that smooth term is removed from the model.

Spline fits	Edf	Smooth effects
PREVENTION MODEL (Nobs = 151, Adjusted R² = 0.449, Deviance explained = 49.1%)		
prevention (yes) ^a	-0.1 ± 0.59	-
s(number of buildings in a 200 meters radius): prevention=no	0.5	no effect
s(number of buildings in a 200 meters radius): prevention=yes	~0	no effect
s(X-coordinate, Y-coordinate)	0.5	no effect

s=spline; Approximate significance of smooth terms based on p-values: 0 *** 0,001 ** 0,01 * 0,05 ^ 0,1 ~0 = values <0.1; ^a linear fit for which is reported the estimate ± standard error instead of the *Edf*

Figure 2. Distribution of the number of apiaries in relation to the density of buildings surrounding the apiaries in a 200-meters radius in areas predicted to be at very low, moderate and high risk of bear damage (classified according to the original household model). The apiaries are classified according to the presence or absence of measures to prevent damage. Note that in the apiaries located in areas predicted to be at moderate or high risk the density of buildings is relatively low. Also note that the majority of apiaries with preventive measures are surrounded by less than three buildings.

In summary, we found that the results of the analyses proposed by the Referee were less accurate and less informative than our analyses and therefore we have left the analyses we

conducted in the previous version of the manuscript. Seeking for clarity and to avoid possible misunderstandings, we have moved the results from the preventive model into a separate table (Table 2) and we have added a new figure in the supplementary materials (Fig. B7).

The approach to scale integration needs to be clarified. It is not clear how you combined the different normalized layers ("based on map overlaying" L 217-218 is unclear). See eg. deCesare et al. 2012 [1], Pitman et al. 2017 [2], Fattebert et al. 2018 [3] for recent examples of scale integration in different taxa.

RESPONSE: Thanks. Seeking for clarity we have revised the suggested articles and modified the text explaining scale integration in more detailed (lines 251-285)

L218-220, Figure 1 and Figure 3: I really wonder what is the benefit of resampling the scale integrated layer at the local and landscape coarser resolution. I would imagine that it is most beneficial for the design of management and conservation intervention to work with a single, scale-integrated map that account for drivers at broad and fine scale. Resampling at coarse resolution defeats the idea of scale integration, in my opinion. I do understand that the output are different from the corresponding single-scale maps, but I would question their benefit for applications.

RESPONSE: Although rescaling may not be needed it can be useful, especially when management decisions are taken at broad scales, as it is often the case regarding the management of human-wildlife conflicts (see [4]). We proved that rescaling the scale-integrated map gives more accurate predictions of damage at large scales than the predictions of single-scale models, and that can help managers to save time and money (lines 505-523). Nevertheless, we understand that this is not the most important result of our study and is more related to applied conservation than to the description of the ecological processes that drive human-wildlife conflicts. Accordingly, we have removed the rescaled maps from figures 1 and 3, leaving a more simplified version of these figures in the main manuscript, and moved the original figure with the rescaled joint probabilities to the Supplementary Materials (Figure B10). We believe that this can also satisfy the Editor's suggestion to focus the discussion of the paper on broader biological implications.

Looking at figure 3, I wonder why you projected risk beyond bear range: you need to justify this or then limit your spatial protection within the blue polygon (Fig. 3). My intuition is that risk should be trivially zero (at all 3 scales) if no bear are present... Unless you consider the risk map as a habitat suitability model and wish to map potential risk if bear were present/when they recolonize, which could inform prevention measures ahead of the recolonization by bear. If this the case, this point needs to be clarified.

RESPONSE: Maybe our explanations were not sufficiently clear. The reasons explained by the Referee were already given in the previous version of the manuscript. We have now elaborated a bit more this justification (lines 252-256) and improved clarity in different parts of the manuscript (lines 339-341, 425-427, 809-811, 823-827).

Also, it is unclear if you have apiaries distribution data beyond the bear range area to be able to project risk over there. You would need apiaries data to be able to inform the grid cells with values of the distance to forest.

RESPONSE: That was also already specified in the previous version of the manuscript. At the landscape scale (5x5 km grid), we estimated the predicted probability of apiary presence as a predictor of damage risk (lines 127-129 and Appendix A in the Supplementary Material). At the

household scale (0.25x0.25 km) we calculated all predictors (including the distance to forest) using the centroid of the cell as a reference point (line 261). We calculated all the predictors within and beyond the bear distribution area and then predicted the probability of damage based on the coefficients of the household model (line 256-261). We hope this is clear enough in the new version.

L223-224 Unclear

RESPONSE: we have deleted that part to reduce the length of the manuscript and left that explanation in the legend of the Figure 3 (lines 829-843).

LITERATURE CITED

1. Decesare NJ *et al.* 2012 Transcending scale dependence in identifying habitat with resource selection functions. *Ecological Applications* **22**, 1068–1083.
2. Fattebert J, Michel V, Scherler P, Naef-daenzer B, Milanesi P, Gruebler MU. 2018 Little owls in big landscapes : Informing conservation using multi-level resource selection functions. *Biological Conservation* **228**, 1–9. (doi:10.1016/j.biocon.2018.09.032)
3. Pitman RT *et al.* 2017 Cats , connectivity and conservation: incorporating data sets and integrating scales for wildlife management. *Journal of Applied Ecology* **54**, 1687–1698. (doi:10.1111/1365-2664.12851)
4. Miller JRB. 2015 Mapping attack hotspots to mitigate human-carnivore conflict: approaches and applications of spatial predation risk modeling. *Biodiversity and Conservation* **24**, 2887–2911. (doi:10.1007/s10531-015-0993-6)

Appendix B

Response to Referees on the revised paper

“The spatial ecology of conflicts:

Unraveling patterns of wildlife damage at multiple scales”

(Ref: RSPB-2021-1394)

**Submitted to:
Proceedings of the Royal Society B**

Dear Editor,

Many thanks for your recent letter inviting us to make a minor revision of our paper *The spatial ecology of conflicts: Unraveling patterns of wildlife damage at multiple scales* (RSPB-2021-1394) before publication. We really appreciate the opportunity to publish the paper in Proceedings B, as well as your time and the time of the two Referees in revising again the manuscript and providing constructive comments. We believe that they have largely helped us to improve the manuscript.

Thank you very much for your time and consideration.

Response to Referees

(PLEASE NOTE THAT LINE NUMBERS REFER TO THE REVISED MANUSCRIPT WITH TRACKED CHANGES)

Associate Editor

The referee commended the authors on the thorough revision to the manuscript, and I also find that the new text and presentation are greatly improved. Please see the remaining comments of the referee, which refer principally to presentation.

RESPONSE: Thank you very much. Please find below the responses to the comments of the Referee 2.

Referee 2

I have read the revised version of the manuscript by Bautista et al. titled “The spatial ecology of conflicts: Unraveling patterns of wildlife damage at multiple scales”. I find this revised version of the manuscript much clearer than the previous one. I commend the Authors for their thorough revision. In particular, I appreciate the effort in exploring my suggestion for modelling risk of apiaries at the household level without using the protected ones in the training data and the detailed presentation of the results in their rebuttal. I have only a few minor comments remaining (line numbers correspond to the unmarked revised version):

RESPONSE: We really appreciate this comment. The Referee’s first revision greatly helped us to improve the manuscript. We have now acknowledged both anonymous referees in the ms (line 439).

L72 you should clearly spell out what risk you are talking about here, before you can use the term ‘risk’ alone: “To evaluate this hypothesis, we used modelled the risk of beehives to be damaged by bears at three scales encompassing...”

RESPONSE: We have rephrased according to the Referee’s suggestion (lines 72-73)

L76 you should mention here the other model at this scale, too (i.e., the preventive model) – you have done more than these 3 models.

RESPONSE: We have added one more sentence explaining briefly the preventive model (lines 78 - 79). We have also specified it in the first sentence of the models’ subsection (lines 182-183) to keep consistency throughout the manuscript.

L164 I suggest that you move this entire paragraph earlier in the Methods, and before the predictors’ section (L116) – you might have to move the rest of the model section (L150), too.

This should inform the reader about what is attempted generally, before diving into the details of each scale.

RESPONSE: We have moved the paragraph that was starting in the line 150 at the beginning of the predictor's section to inform about the main objective of our analyses (lines 120 – 132). We have also considered moving the models' specifications (paragraph starting in the line 180 -"L164" in the referee's comment-) before the predictors section but we have decided not to do so because we are afraid that the order of the Methods section could become counterintuitive for many readers; in most journals (including Proceedings B) the calculation of the predictors is most of the times explained before the description of the models, and not after. If possible, we would prefer not to follow that referee's suggestion and keep the typical Method's order to avoid disrupting the typical flow of the manuscript.

Also, one thing that remain unclear, is how the 123 damaged vs the 170 non-damaged occurrence are treated as a response variable in your GAMs – you need to explicitly tell a relative naïve reader how you have coded them, and which are 1's and which are 0's.

RESPONSE: We have added one sentence clarifying how we coded each cell and apiary to assess the risk of damage (lines 130 -132). We have also explicitly explained that we used damage occurrence versus absences as response variable in our models (lines 184 -185).

L216 you need at least one reference to back up this approach, e.g. DeCesare et al. 2012. I suggest that you rephrase as: "...by multiplying the damage probabilities at the landscape, local and household scales, following DeCesare et al. 2012."

RESPONSE: We have included the reference as suggested by the referee (line 238) and updated the reference list.

LITERATURE CITED

1. Decesare NJ *et al.* 2012 Transcending scale dependence in identifying habitat with resource selection functions. *Ecological Applications* **22**, 1068–1083.

Reducing the extent of the manuscript

In order to fulfill the 10-pages limit set by the journal, we have reduced the length of the manuscript by 772 words. To do so, we have removed 12 references and deleted or modified the sentences in the lines 10, 11, 38, 42, 43, 44, 46, 54, 63-66, 106, 144, 145, 148-150, 230-233, 266, 270 and 362-369.